# The Algorithm for Assessing the Effects of Distance Education in General Education on Students' Academic Results

Vaidas Gaidelys [1] , Rūta Čiutienė [1,*] , Gintautas Cibulskas [2] and Asta Baliute [1]

[1] School of Economics and Business, Kaunas University of Technology, LT-44239 Kaunas, Lithuania; vaidas.gaidelys@ktu.lt (V.G.); asta.baliute@ktu.lt (A.B.)
[2] Faculty of Social Sciences, Arts and Humanities, Kaunas University of Technology, LT-44239 Kaunas, Lithuania; gintautas.cibulskas@ktu.lt
* Correspondence: ruta.ciutiene@ktu.lt

**Abstract:** Distance education has gained great interest during the COVID-19 pandemic, when schools all over the world faced the challenge of transferring the traditional education processes to digital platforms. In this context, school communities not only discovered the new opportunities, but also encountered a number of problems when trying to provide high quality distance education and minimise learning losses for students. The rapid transition to distance education has had a negative impact on the academic results and daily routines of the students from all social groups, especially socially vulnerable families. Thus, it has become relevant to research various aspects of students' learning losses. The purpose of this article is to theoretically substantiate the algorithm intended for assessing the effects of distance education on students' academic results. The research was based on the methods of scientific literature analysis, secondary data analysis, theoretical modelling, inductive content analysis, and expert evaluation. The algorithm is intended for general education institutions. Results: After conducting scientific literature and secondary data analysis, the theoretical model for assessing the effects of distance education on students' academic results was developed and substantiated. The model consists of four structural parts, represented by 11 criteria, which reveal the effects of distance education in general education following the appropriate algorithm.

**Keywords:** distance education; learning losses; theoretical assessment model; algorithm

## 1. Introduction

Recently, with the acceleration of the digitization process, distance learning is gaining an increasing role in education. Although distance education has been applied in higher education and non-formal education for a long time, it has been little spread or in most cases not applied at all in general education institutions. Distance learning/education has gained great interest during the COVID-19 pandemic which caused the greatest disruption of the education and training processes worldwide [1]. Distance education is defined "as an education system based on the Internet and interactive technologies to enable teachers and students from different locations to meet in real time" [2]. Schools all over the world faced the challenge of transferring the traditional education processes to digital platforms in a short period of time. In this context, school communities not only discovered the new opportunities of transforming the traditional education processes, but also encountered a number of problems when trying to provide high quality distance education and minimise learning losses for students.

Different countries have different experiences in organising distance education. Previous studies, focused on the effects of distance education, reveal that the very great difficulties were encountered by developing countries. Many previous studies highlight the negative effects of distance education not only on academic results, but also on the daily routines of students and their family members. The OECD [3] and UNESCO [4,5] analyse the negative effects of school closures on various aspects of students' lives, including their

health condition. The UNESCO [5] focuses on the special educational needs of the disabled students and emphasises the risk of exclusion and other social difficulties, faced by this group of students.

When implementing the project "Distance education of children during the COVID-19 pandemic: threats and opportunities from an ecosystem point of view: e-children", the researchers from Vilnius University (VU) addressed the effects of distance education on the physical and mental development of the preschoolers and 1–8 grade students [6]. The research finds that distance education has had a negative impact on the students' emotional state and physical health and has led to greater social isolation and lower motivation to study.

According to the UNESCO [4], the rapid transition to distance education has had a negative impact on the academic results and daily routines of the students from all social groups, especially socially vulnerable families. The European Foundation for the Improvement of Living and Working Conditions emphasises the economic problems: higher monetary and time costs of childcare and education. Some other studies tend to focus on the positive aspects of distance education [7].

Few previous studies, however, address the issues of assessing the knowledge gaps of the students after their returning to schools, developing the learning loss compensation mechanisms, and preventing occurrence of the similar learning losses in the future. The adequate assessment of the knowledge gaps must be based on a reliable theoretical model and an appropriate algorithm. Then it can effectively contribute to knowledge improvement.

The major purpose of this article is to theoretically substantiate the algorithm intended for assessing the effects of distance education on students' academic results. In the paper, the algorithm is understood as "systematic procedure that produces—in a finite number of steps—the answer to a question or the solution of a problem" [8].

The research, introduced in this article, was based on the methods of scientific literature analysis, inductive content analysis, and expert evaluation. It helped to develop and theoretically substantiate the model intended for assessing the effects of distance education on students' academic results. The model consists of four structural parts, represented by 11 criteria, which, in their turn, are assessed by employing the appropriate algorithm.

## 2. Literature review

### 2.1. The Negative Effects of Distance Education

Most previous studies tend to emphasise the negative effects of distance education on students' learning and academic results. Some authors focus on lower motivation and its determinants. For instance, Seaman et al. [8] analysed the impact of distance learning on the results obtained in the higher education faculty, operating in the STEM area. Their study finds that most of the respondents had no experience of distance learning before the COVID-19 pandemic. The lack of motivation and academic integrity in the process of distance education were identified as the major problems. In addition, the study reveals the importance of the resources, such as computer technologies, a reliable Internet connection, etc., on the quality of distance education [9].

Skliarova et al. [10] conducted the study on the online teaching strategies appropriate for blended and face-to-face higher STEM (science, technology, engineering, and mathematics) education at the university of Aveiro. The researchers note that a properly selected teaching strategy leads to greater students' motivation and engagement, helps to build the positive relationship with students, and contributes to teacher and student satisfaction. The results of the empirical research also indicate that most of the respondents prefer studying in a physical place rather than at a distance; it is difficult to motivate students, ensure the feedback, and be in touch with students and colleagues. The teachers emphasise the importance of extra time: they need more time to properly prepare for distance teaching. The problem of unethical behaviour is also noticed.

Skliarova et al. [11] researched the online teaching and learning practices from a students' perspective. Their research finds that students' motivation and engagement

are greatly affected by the study environment: innovative, interactive, and attractive methods of distance education, shorter lessons, and more practical tasks. It should be noted that distance education is also associated with challenges: primarily, a loss of students' concentration, a lack of a personal contact, and technical issues.

Coman et al. [12] analysed the online teaching and learning from a students' perspective in higher education during the COVID-19 pandemic. The researchers collected and analysed the answers provided by 762 students at two largest Romanian universities. They found that technical issues are the key challenge, followed by the problem of teachers lacking technical skills for distance education, and inappropriate teaching methods which are not adapted to the digital environment. The poor communication with teachers, as well as the lack of this communication, were categorised as least significant.

Futterer et al. [13] investigated the components of the effective distance teaching, and analysed how it is affected by technology familiarity at secondary schools in Germany. The authors examined 729 reports, provided by 729 ninth-grade students who were enrolled for two subjects (mathematics and German). The findings revealed that merely being acquainted with technological tools is insufficient, and it is crucial to comprehend the benefits of technology in enhancing pedagogy in a virtual classroom, as is done in traditional face-to-face classrooms.

Vit [14] investigated the impact of the pandemic-induced distance learning on the capacity of educational institutions to prevent learning losses in Hungary.

Distance education led to a decrease in students' physical activity, communication with peers, and a lack of personal space at home. Students tended to spend longer hours at their computers both during their lessons and free time, which could have led to sleep disorders and other mental health problems [15].

Cappelle et al. [16] focused on the effectiveness of distance learning in India during the COVID-19 pandemic. They investigated which technologies and distance learning methods were used after the school closures, and what effects they had on students and their parents. The analysis revealed that the perception of learning was affected by three major factors: the frequency of a teacher contact, the learning modality, and utilisation of the students' free time. The author also notes the role of the state policies. They argue that the clear standard procedures for managing the digital content, as well as the effective implementation of these procedures, can positively affect the process of distance learning/teaching. The study provides the rapid assessment, based on the telephone survey.

Bonal et al. [17] investigated the impact of the school closures on the learning gap observed between the children from different social backgrounds in Catalonia. The study was based on 35,419 responses, collected through an online survey which was administered to families with children aged 3–18 from 26 March to 30 March 2020. The study revealed that the children from different social backgrounds had different educational potential: the children from socially disadvantaged families had fewer opportunities for learning and extracurricular activities, while the children from the families with higher economic, social, and cultural capital tended to participate in the activities which could be transferred to the digital space (e.g., artistic activities, learning foreign languages, etc.). Nevertheless, the children from the families with lower economic, social, and cultural capital were more likely to engage in sports. When assessing how often parents tended to stop extracurricular activities for their children, the authors found that the families with lower economic capital were more likely to do that because of the higher costs.

Vidergor [18] researched how teachers' innovativeness affected their distance teaching autonomy, accountability, and practices. A total of 200 teachers, representing elementary and secondary schools in Israel, participated in the survey, and were asked to fill in a questionnaire. The results revealed that teachers' work experience directly affected their self-innovativeness, with older and more experienced teachers perceiving themselves as more innovative in distance education, compared to less experienced colleagues. The second finding was that professional development tended to affect distance education practices but had no effect on teachers' innovativeness. The study proposes that the increasing

innovativeness of teachers can positively affect their autonomy and accountability and contribute to more high-quality distance education practices. Thus, the authors recommend directing the programs of professional development towards promoting teachers' innovativeness and creating new combinations for hybrid teaching.

Maatuk et al. [19] found that the quality and effectiveness of distance education are significantly affected by the technical and financial support, staff training, better working conditions, the technological basis, skills, copyright protection, and professional development. Their results also indicate that distance education tends to increase the workload, especially for learners. Although distance education requires additional financial and material resources, some positive aspects, such as improving computer literacy, can also be envisaged. For successful distance education, it is necessary to pay attention to the different basic preparation and motivation of the learners.

Jusienė et al. [20] argued that distance education led to many challenges in terms of academic ethics because teachers hardly had any tools/methods to control their students. The tasks were not adapted to the mode of distance education, which made it easier for students to cheat. The academic ethics depended on the integrity of students, the attitudes of their parents, and the ability of teachers to lead and control their classes.

Klisovska et al. [21] found that distance education posed many challenges to students because most of the work had to be done on the computer, while previously it had been done in notebooks. Thus, students needed some time to get used to the new system.

Summarising, the literature analysis revealed the following negative aspects of distance education (see Table 1):

**Table 1.** Negative aspects of distance education.

| Number | Negative Aspects of Distance Education |
|:---:|:---:|
| 1 | Social inequality: children from social risk families had less potential for learning because they were lacking study-favourable conditions and full meals which had been provided in schools. |
| 2 | Lower motivation: not being motivated directly, students lost their motivation to study. |
| 3 | Increased workload for teachers and stress caused by innovations: when schools were closed unexpectedly and for an indefinite period, teachers were confused on how to perform their functions properly, how to communicate, and how to provide educational assistance to students. The transition to distance education platforms was often incoherent and stressful. |
| 4 | Parents were not ready for distance and home education: when schools were closed, parents were often asked to help their children study at home, but not all parents have the appropriate knowledge, skills, and resources. |
| 5 | Challenges in creating, maintaining, and improving distance education: the need for distance education emerged unexpectedly, and the current distance learning/teaching platforms had not been adjusted to the intensive use. The emergency transition from class to home education posed challenges to both people (teachers, students) and technologies (the increased loads for the Internet and computer equipment). |
| 6 | Poorer childcare: having no other choice, working parents tended to leave their children alone, which might have led to risky behaviour, including greater peer exposure and substance use. |
| 7 | Greater economic losses: working parents often tended not to go to work to be able to take care of their children, which resulted in wage losses and/or negative productivity changes. |
| 8 | Social isolation: schools are places of social activity and human interaction. When they are closed, many children and young people lack the social contact, which is essential for learning and development. |

**Table 1.** *Cont.*

| Number | Negative Aspects of Distance Education |
|--------|----------------------------------------|
| 9 | Challenges in evaluating academic results: planning of the tests and exams, which are followed by the transition to a higher grade or a higher study level, calls for innovative solutions. It is not uncommon to postpone or cancel an exam. Disruptions in the evaluation process are stressful for students and their families and reduce the motivation to study. |
| 10 | Academic ethics issues: distance education posed many challenges in terms of the academic ethics; teachers hardly had any tools/methods to control their students. The tasks were not adapted to the mode of distance education, which made it easier for students to cheat. The academic ethics depended on the integrity of students, the attitudes of their parents, and the ability of teachers to lead and control their classes. |

Other studies, however, highlight the positive aspects of distance education.

*2.2. The Positive Effects of Distance Education*

Distance education became a necessary alternative to traditional education during the COVID-19 pandemic. According to Smith [22], distance education offered certain advantages in this unusual period. Above all, distance education ensured the continuity of studying. Brown [23] argued that even if schools were physically closed, and students were isolated in their homes, distance education provided an opportunity to continue education, which helped to prevent a major break in education and supported the academic progress. In addition, distance education contributed to the use of digital technologies and digital literacy. Johnson [24] argued that this crisis forced schools and students to quickly adapt to the new reality and acquire the necessary digital skills. This can provide the long-term benefits as digital competences are becoming increasingly important nowadays.

Jusienė and Būdienė et al. [20] found that distance education:

- reduced fatigue and the somatic symptoms, which are related to getting up early in the morning, a busy schedule, and the school stress, in some children (especially in grades 5–8);
- improved the digital competences of all children;
- involved parents in the education of primary school students;
- allowed parents to get acquainted with the abilities and characteristics of their children in primary schools and pro-gymnasiums;
- improved parents-children relationship and children's well-being, if parents experienced less work-related stress and had lower workload;
- created the natural preconditions for the development of teachers' general and professional skills, in particular, in the areas of information and computer technologies, communication, self-management, and languages;
- forced teachers to refine and update the educational content and process with consideration of the necessity to shorten the screen time, availability of the new forms of resources and information, students' digital competences, and the increased need for cooperation;
- facilitated the individualisation and differentiation of education, when different contact time was set for the motivated students with high learning potential, and the students with lower achievements and/or special educational needs who needed individual assistance;
- contributed to the renewal of the technical tools which could be used for both distance and face-to-face education;
- accelerated the supply of students and (partially) teachers with technical equipment for distance learning/teaching;
- allowed to test different distance learning/teaching platforms which could be used in the future;

- strengthened the cooperation between the school and the family—parents were involved in children's education, common challenges shaped the positive attitudes of school administrators, teachers, and parents to mutual communication;
- highlighted the importance of a teacher's work, and thus stimulated the respect for a teacher's profession.

The study conducted by Gaidelys et al. [25] revealed the following positive aspects of distance education: when studying at home, students feel more comfortable, they can get the necessary help, record lessons, and review them later, and select a quiet and comfortable environment for studying. The authors also note that a safe home environment created the preconditions for students to feel psychologically calmer and safer.

Distance education provides flexibility for students and teachers. As noted by Johnson [24], online classes allow students to learn at their own pace and reduce transportation and lost time costs. Teachers can adapt teaching materials to the needs of different students and use various interactive tools which stimulate students' involvement in the teaching/learning process.

Distance education can open new learning/teaching opportunities. According to the research by White et al. [26], virtual environments and learning/teaching platforms can provide access to a variety of educational resources regardless of a geographic location. This allows students to get to know different cultures and languages.

Klisowska [21] summarised that distance education is an excellent form of learning/teaching because it possesses many advantages, such as time saving and access to various materials, but it requires teachers' dedication to encourage students to expand their knowledge, as well as students' self-control and motivation.

Distance education can become an effective educational model on the condition that the appropriate infrastructure, staff training, and flexible structures that facilitate decision making are developed even after the pandemic and informal communication channels are available [24,27–30].

Summarising, the literature analysis revealed the following positive aspects of distance education (see Table 2):

**Table 2.** Positive aspects of distance education.

| Number | Positive Aspects of Distance Education |
|---|---|
| 1 | Continuity of education: distance education allows to continue studying regardless of one's location. |
| 2 | Improved digital literacy and IT competences: students and teachers were forced to improve their skills and acquire new competences. |
| 3 | Reduced fatigue: students did not have to get up early, or travel (sometimes long distances) to school. |
| 4 | Parents were involved in their children's learning process: after schools were closed, parents were often asked to help their children study at home, and they became more familiar with the educational process. |
| 5 | Improved communication between students and parents: parents tended to spend more time with their children and assisted them. |
| 6 | Teachers renewed their material presentation skills: having no other choice, teachers tended to look for innovative ways to present the material in a more interesting and targeted manner. |
| 7 | Improved personalisation and differentiation. Teachers discovered new ways to personalise teaching. |
| 8 | Renewal of the IT base: schools were forced to renew their current IT resources, acquire the new equipment, and supply it to the teachers and students. |

**Table 2.** *Cont.*

| Number | Positive Aspects of Distance Education |
|---|---|
| 9 | Improved cooperation between the school and the family: parents were forced to communicate with teachers and school administration more actively. |
| 10 | Increased respect to a teacher's profession: parents could observe the teaching process and see the teachers' effort. |

On balance, the literature analysis proposes that distance education has had both positive and negative effects. Thus, it would be relevant to have a comprehensive instrument which would allow to assess the multi-sided consequences of distance education.

## 3. Assessment of the Effects of Distance Education

To assess the consequences of distance education, it is necessary to have reasonable criteria. Previous studies cover different aspects which affect organising and conducting distance education, and which can serve as prerequisites for selecting the assessment criteria.

Having conducted the analysis of distance learning in China during the outbreak of COVID-19, Huang et al. [31] found that distance learning/teaching was based on the strategy of distance education. The strategy covers six major aspects: (a) infrastructure, (b) education tools, (c) education resources, (d) teaching and learning methods, (e) services to teachers and students, and (f) cooperation between schools and governments.

Timmons et al. [32] focused on the uncommon challenges which were related to distance education in kindergartens and primary schools. Their data analysis disclosed five major aspects which significantly affected students' academic results: equal opportunities, synchronous and asynchronous teaching and learning, social and emotional effects on students, the impact on academic achievements, and the impact on parents/families.

Having analysed the SELFIE self-analysis tool and the results of previous studies, including the latest foreign findings, Daukšienė et al. [33] detected the following areas where adjustments must take place to ensure the effective distance education: 1. Strategy, management and administration; 2. Information technology (IT) infrastructure; 3. Digital content of teaching/learning; 4. Digital competences and continuous professional development; 5. Learning, teaching and assessment in the digital space; 6. Support system for students and teachers; 7. Partnership, cooperation and networking; and 8. Quality assurance.

Gaidelys et al. [25] identified the following factors which affected distance education: IT and infrastructure; teaching/learning and assessment; assistance to teachers and students; educational content; cooperation; physical and mental health; involvement of students; digital competences; study environment; and organisation.

The literature analysis allowed to draw four blocks (learning/teaching outcomes and results; process; infrastructure and assistance tools; and effects on one's physical and mental health), which integrate the major factors, proposed by the relevant scientific literature, into a logical scheme for assessing the effects of distance education (see Figure 1).

This scheme served as a basis for developing the research algorithm.

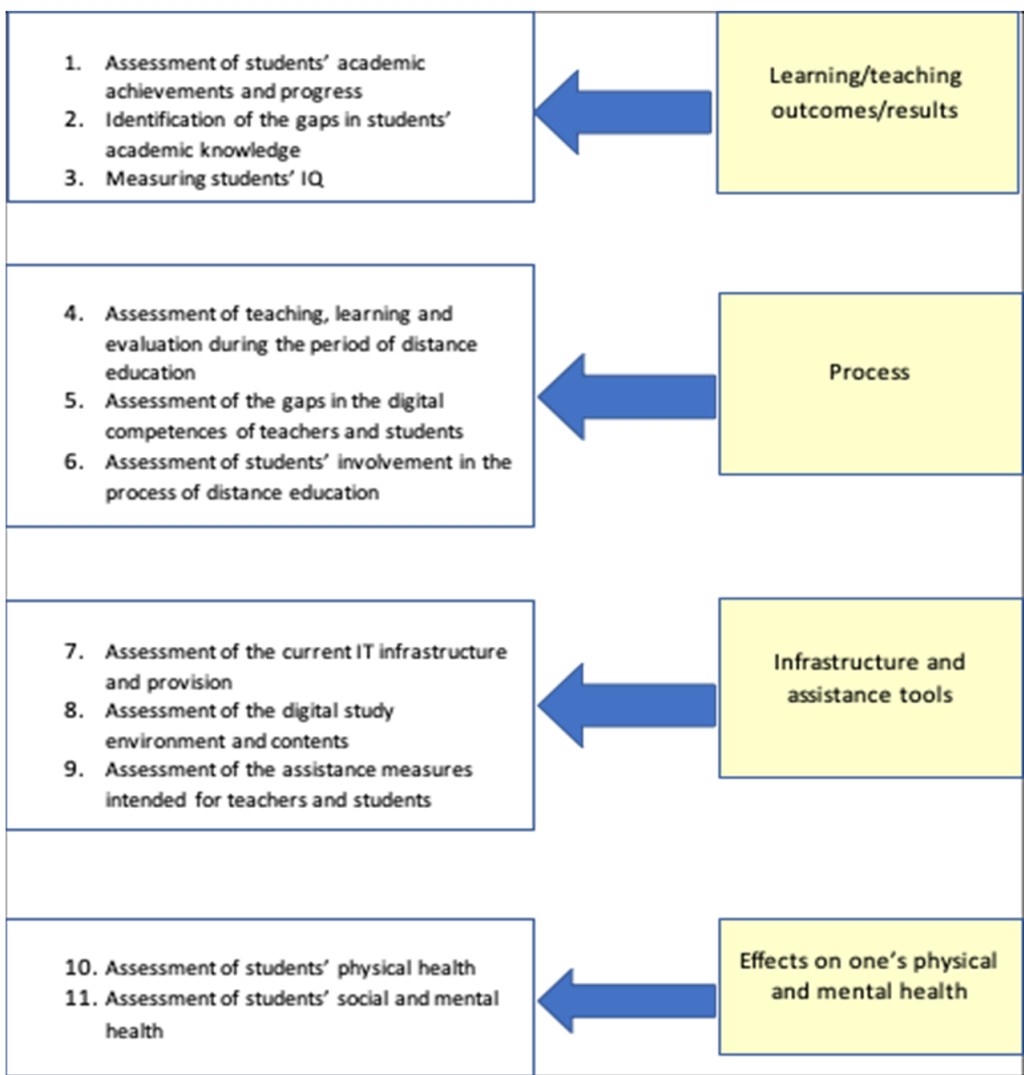

**Figure 1.** Major factors of assessing the effects of distance education.

## 4. Methodological Approach

This section reviews the methodology selected for the research, and explains each stage of the modelling, which was based on the results of previous studies.

The major stages of the research process are presented in Figure 2.

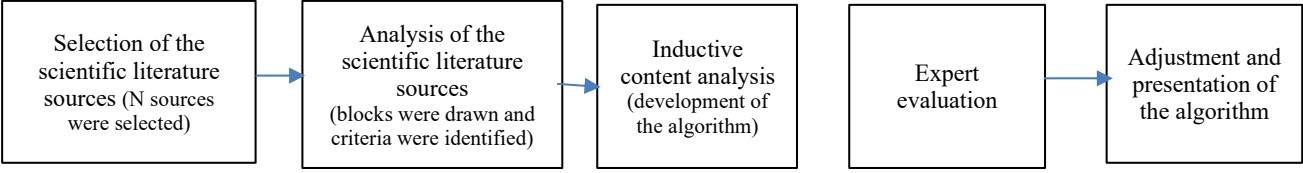

**Figure 2.** Major stages of the research process.

The blocks and linear arrows represent the research process, while the result of a stage is presented in parentheses. The major stages of the research process are described below.

The relevant scientific sources were selected from the scientific research databases (SCOPUS and Google Scholar). The selection was conducted by employing key words and search filters.

The systematic review of the scientific literature is defined as a systematic literature review as a systematic analysis of scientific literature, when the most relevant scientific studies are selected with consideration of the predetermined criteria. The analysis of

scientific articles was the main method of data collection in this study. As mentioned above, the selection of the relevant articles was conducted in the SCOPUS and Google Scholar databases. These databases provide the reliable, high-quality research results, and are rich in the relevant scientific articles.

*Research Procedures*

A bibliographic search in the SCOPUS database was conducted in February 2023. The search was based on the following descriptors: Education (title), Distance learning (title, abstract, keywords), Issues (title, abstract, keywords), Positive and negative aspects of distance education (title, abstract, keywords). The articles had to be: (1) in English, (2) published in a peer-reviewed scholarly journal, and (3) in open access. The empirical studies had to be conducted in the period 2020–2023.

After the automatic data selection in the SCOPUS database, the selected articles were saved in Excel, where they were assigned serial numbers. Following the procedures, 35 sources, relevant for further analysis, were selected. The reports of the relevant scientific research projects and the reviews provided by international organisations, such as the OECD and UNESCO, were also selected for the analysis (they were extracted by employing Google Search engine).

The method of text analysis was used for identifying the relevant criteria, which were categorised as the positive or negative aspects of distance education.

The third stage covered the inductive content analysis, which allowed to integrate the structural parts of the algorithm. Induction (Lat. *inductio*—introduction; English—*induction*; German—*induktion*) refers to a way of thinking, when partial statements, cases, and features are taken into account to make a generalisation, and when the specific knowledge and individual facts are considered to make general conclusions. Induction is one of the methods of generalisation [34]. In the social sciences, induction is a method of inquiry, whereby general statements or theories are derived from the research material. Induction begins with specific observation and data collection, then particular theories or more general propositions are derived from the findings, observations, and data analysis. The major advantage of the inductive method is that it allows the derivation of new propositions or theories which are based on the specific data or observations.

In the fourth stage, the developed algorithm was submitted for expert evaluation. A total of 3 experts, representing different management levels in the Lithuanian education system, were selected for the expert evaluation by applying the criteria-based selection. The experts had to meet the predefined criteria and represented different areas: 1 expert represented the area of the municipal education policy formation, 1 expert was the headmaster of the gymnasium which runs primary, basic, and secondary education programs, 1 expert was the headmaster of the gymnasium which runs secondary education programs, and 1 expert was the headmaster of a primary school. A theoretical algorithm, developed on the basis of the scientific literature analysis, was submitted for the expert evaluation. For each part of the algorithm, the experts had to provide their observations concerning the practical applicability of the algorithm. Considering the remarks of the experts, the authors of this research adjusted the algorithm.

## 5. The Results of the Research

The scientific literature analysis, the logical scheme for assessing the effects of distance education, and the expert evaluation allowed to develop the algorithm for assessing the effects of distance education on students' academic results. The algorithm considers the effects of distance education on the learning outcomes/results, process, infrastructure and assistance tools, and students' physical and mental health. The algorithm comprises 11 steps (see Figure 3).

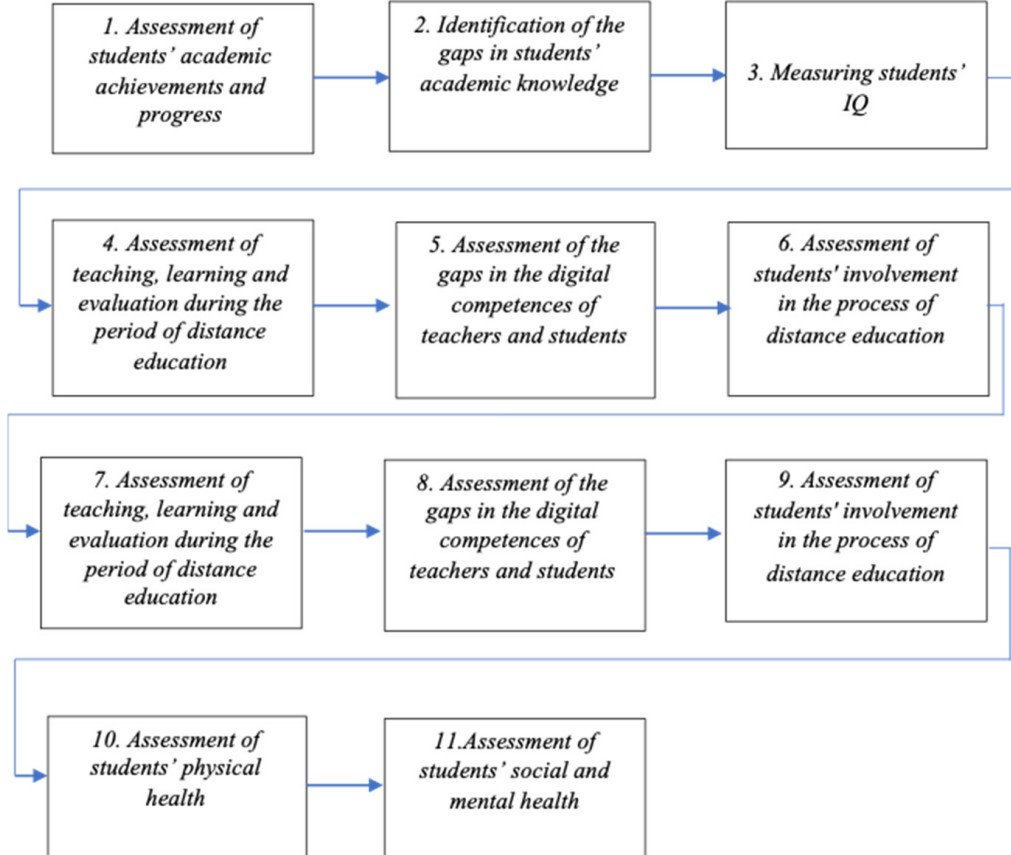

**Figure 3.** The algorithm for assessing the effects of distance education on students' academic results.

The area of the learning outcomes/results covers:

1.  Assessment of students' academic achievements and progress. The algorithm allows to assess students' academic achievements at the current moment, i.e., it considers the academic results after distance education or during it. Then, the current results are compared to the ones in previous periods. The changes in the study load during the period of distance education are also considered.

2.  Identification of the gaps in students' academic knowledge. After assessing students' academic achievements and progress, the gaps in their academic knowledge are identified. The subject knowledge tests are used for this purpose.

3.  Measuring students' IQ. The Wechsler Abbreviated Scale of Intelligence (WASI) or a similar method are applied to measure students' IQ which is compared to the results of previous tests.

The area of the process covers:

4.  Assessment of teaching, learning, and evaluation during the period of distance education. The algorithm allows to assess how teaching, learning, and evaluation processes are organised and supported during the periods of distance education. The assurance of objectivity and fairness is also considered.

5.  Assessment of the gaps in the digital competences of teachers and students. Digital competences play an essential role during the periods of distance education, and can, therefore, be recognised as one of the prerequisites for successful distance education. It is important that both teachers and students are properly prepared and able to use IT. This step allows to assess the gaps in their digital competences, and reveals which competences are insufficient and need to be improved.

6.  Assessment of students' involvement in the process of distance education. Distance education requires significantly more awareness, motivation, and full involvement of

students. On the one hand, it is important to apply inclusive teaching methods, while on the other hand, students' self-awareness and awareness need to be improved. In both cases, it is necessary to understand which factors can promote involvement in the process of distance education.

The area of the infrastructure and assistance tools covers:

7. Assessment of the current IT infrastructure and provision. Infrastructure and provision are another important factor in ensuring the smooth process of distance education. Therefore, it is necessary to assess the current and the necessary IT infrastructure. The assessment should be performed from two perspectives: what IT infrastructure is required by teachers and students, respectively.

8. Assessment of the digital study environment and contents. The algorithm considers the current study environment and allows to assess how it meets the expectations of students and teachers, how the teaching material is presented, and whether it is sufficient to achieve the desired learning/teaching outcomes.

9. Assessment of the assistance measures intended for teachers and students. The algorithm allows to assess what assistance (if any) is provided to teachers and students during the process of distance education.

The area of the students' physical and mental health covers:

10. Assessment of students' physical health. The algorithm allows to assess how students' physical activities changed during the process of distance education, and what changes in students' physical condition occurred as a result.

11. Assessment of students' social and mental health. Distance education led to a reduction in students' social contacts and physical activities, which had a detrimental effect on their social and mental health. Thus, the assessment of students' social and mental condition would help identify whether the professional counselling and treatment are appropriate.

It is clear that education will not return to pre-pandemic levels. Distance learning is becoming an integral part of education. However, effective distance learning requires close cooperation and involvement of all stakeholders [35,36].

## 6. Discussion and Conclusions

This analysis yields significant findings and carries both theoretical and practical implications concerning the effects of distance education as part of general education on students' academic results. Even though distance education is not a novelty in the general education system, it became a necessity in general education institutions during the COVID-19 pandemic. Neither schools, students, and their parents nor educational decision makers were adequately prepared for the mode of distance education in general education institutions. In most cases, this has led to negative consequences, though some positive effects can also be envisaged. Only with a clear understanding of the negative consequences, one can look for solutions to eliminate the deficiencies and properly prepare for distance education in the system of general education. Additionally, the factors that generate the positive effects should be exploited and stimulated.

The theoretical part of this paper highlights the positive and negative aspects of distance education. Distance education has opened up countless opportunities for learning, and digitization of the educational content has become the inevitable future. Digital education ensures the continuity of education when the traditional face-to-face lessons cannot be conducted. During the COVID-19 pandemic, distance education not only ensured the continuity of education, but also forced the participants of the educational process to improve their digital competences, develop the IT infrastructure, and acquire the tools necessary for studying/teaching online. In addition, distance education revealed the significance of a teacher's role, as well as the importance of the immediate cooperation among parents, students, and educational institutions.

Despite the positive effects of distance education, it has deepened the social and economic inequality, decreased students' motivation, increased teachers' workload, raised the issues of the evaluation of students' progress and academic ethics, revealed parents' unpreparedness to help their children with studies, etc. The abovementioned positive and negative aspects of distance education have affected students' academic results. In many cases, the results of distance education are worse than those of face-to-face learning. Considering this, it is necessary to understand the real consequences of distance education and develop the methods to improve the situation.

This article introduces the algorithm for assessing the effects of distance education on students' academic results. The algorithm was developed on the basis of the findings presented in previous scientific studies. It considers the effects of distance education on the learning outcomes/results, process, infrastructure and assistance tools, and students' physical and mental health.

The mode of distance education in the system of general education during the COVID-19 pandemic is associated with reduced academic motivation [8–11], the lack of digital competences of both students and teachers [13], and insufficient supply of the necessary tools [16], which led to the changes in the common learning outcomes/results. The adequate assessment of the learning gaps would allow to identify individual knowledge gaps in a specific subject and develop the appropriate measures to eliminate these gaps.

Distance education is fundamentally transforming the processes of teaching and learning, including the assessment of students' academic knowledge [31]. It is no longer possible to achieve good results with conventional methods adjusted to the mode of distance education. The processes under transformation [33], digitalisation of the educational content [24,26,27], as well as the integration of information technologies not only require digital skills of teachers and students [20,24], but also tend to change the behaviour of the participants of the educational process [32] in the digital space. The adequate assessment of the process would provide the prerequisites for understanding which factors and tools allow to ensure the high-quality involvement of students in distance education.

The assessment of the infrastructure and assistance tools would allow to see what measures are needed to ensure the smooth process of teaching and learning in the case of distance education. The appropriate infrastructure, assistance services to teachers and students, as well as close cooperation between schools and governments [26,31] facilitate distance education, increase its accessibility, and promote the development of digital skills. The support provided by the administration of schools and national governments would allow to switch to the mode of distance education with the least possible losses.

Physical and mental health is another significant aspect in the context of distance education. Spending more time in the digital space, students devote less time to physical exercises [15], communication with friends, and/or extracurricular activities, which leads to major social and emotional problems [10,23] and physical health disorders [25]. Thus, the assessment of students' physical and mental health could help find the activities which would contribute to its improvement.

Further research could cover a wider audience and the algorithm could be evaluated not only by Lithuanian, but also by foreign experts. In addition, it should be noted that after assessing the effects of the mode of distance education in the system of general education on students' academic results, the measures to improve the situation must be developed.

**Author Contributions:** Conceptualization, V.G., R.Č. and G.C.; methodology, V.G., R.Č. and G.C.; formal analysis, V.G., R.Č. and G.C. investigation, V.G., R.Č. and G.C.; data curation, A.B.; writing—original draft preparation, R.Č. and G.C.; writing—review and editing, R.Č. and G.C.; visualization, V.G. and A.B.; supervision, V.G., R.Č. and G.C.; project administration, V.G.; funding acquisition, V.G. All authors have read and agreed to the published version of the manuscript.

**Funding:** This project has received funding from European Regional Development Fund 13.1.1-LMT-K-718-05-0013 under grant agreement with the Research Council of Lithuania (LMTLT). Funded as European Union's measure in response to COVID-19 pandemic.

**Institutional Review Board Statement:** Not applicable.

**Informed Consent Statement:** Not applicable.

**Data Availability Statement:** The data used for the study is available upon reasonable request.

**Conflicts of Interest:** The authors declare no conflict of interest.

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
