# Peer review of "The Algorithm for Assessing the Effects of Distance Education in General Education on Students’ Academic Results"

_education, doi:10.3390/educsci13090957_

Round 1

Reviewer 1 Report

The pandemic was a starting point for supporting/developing online educational activities to a different level than previously known. Like it or not, this moment has changed the way we perceive education. The way in which teachers have adapted to the training of learners' skills through online educational activities has been studied by many authors, revealing both positive and negative aspects. The authors have managed to capture these aspects by highlighting them very well in the two tables, according to them, represents the method of text analysis that was used for identifying the relevant criteria.

I carefully read the paper and found that the topic of the paper is important, and the paper is written with interest. However, I have a few unclear points that came up after reading it: the methodological approach and the research methodology as well as the other aspects of the research results seem general to me. From what I have read, it does not appear where the study was carried out, but also other data in this regard that would allow the researcher to somehow locate what was studied.

As for the results of the research, again, I can say that the aspects presented are general, all of which are based on the study of the literature and, probably, on the few recommendations given by the experts listed in the paper (in point 4).

Therefore, I consider that the paper needs to be improved, I suggest including that the authors briefly discuss the following related and important issues in the conclusion section or discussion section (the authors may create a discussion section).

Some words are written with other letters than it is known (z instead of s, for example)

Author Response

Dear Reviewer, 

thank you for your comments and remarks for paper improvement. Your comments have been taken into account:

  1. Comment "From what I have read, it does not appear where the study was carried out, but also other data in this regard that would allow the researcher to somehow locate what was studied".  Added, lines 345 - 347.
  2. Comment "...I suggest including that the authors briefly discuss the following related and important issues in the conclusion section or discussion section (the authors may create a discussion section".  Discussion was included into Conclusions section. 
  3. Comment "Some words are written with other letters than it is known (z instead of s, for example)". Corrected. 

Reviewer 2 Report

The paper deals with a very interesting topic, a review of scientific articles was made and the advantages and disadvantages of distance education were clearly presented. The criteria and algorithm for evaluating the effect of distance education on students' academic results are also clearly presented.

The literature should be cited according to the journal's propositions, so inspect and repair that part.

Author Response

Dear Reviewer, 

thank you for your comments and remarks for paper improvement. Your remarks have been taken into consideration. The literature citation was corrected. 

Thank you for your input.

Round 2

Reviewer 1 Report

I have to  congratulate the authors for this new form of the paper.